# Evaluating the Unseen: A Novel Framework for Assessing Unsupervised Concept Bottleneck Models

## Abstract

In recent years, the field of explainable artificial intelligence (XAI) has gained significant traction, with concept bottleneck models (CBMs) emerging as a promising approach to enhance the interpretability of machine learning systems. However, CBMs often rely on expert-annotated concepts, which can be costly and time-consuming to acquire. To address this limitation, unsupervised and label-free CBMs have been proposed, but these come with their own challenges, particularly in assessing the reliability and accuracy of the generated concepts without ground-truth labels. This paper introduces a comprehensive evaluation framework designed to assess the quality of explanations produced by unsupervised CBMs. Our framework comprises a set of novel metrics that evaluate various aspects of the concept outputs, including their relevance, consistency, and informativeness. We demonstrate the effectiveness of our metrics through a series of experiments, showing certain positive correlations between our scores and both LLM evaluations and human judgments. Our work not only fills a critical gap in the evaluation of unsupervised CBMs but also provides a solid foundation for further research into more transparent and trustworthy AI systems.

## 1 Introduction

The quest for explainable artificial intelligence (XAI) has led to the development of concept bottleneck models (CBMs), which strive to enhance transparency and interpretability in complex machine learning (ML) systems. CBMs rely on human-interpretable concepts to mediate predictions, yet their reliance on manually annotated concepts poses a significant bottleneck, limiting scalability and applicability Lai et al. (2023).

Recent advancements in unsupervised and label-free CBMs have attempted to overcome this challenge by automatically extracting concepts from data Oikarinen et al. (2023); Yuksekgonul et al. (2023). While these methods alleviate the annotation burden, they introduce new evaluation complexities. The absence of ground-truth labels makes it difficult to assess the quality and relevance of the learned concepts, and even when labels are available, they may not align perfectly with the unsupervised outputs due to distributional differences and inherent gaps between labeled and unlabeled data Wei et al. (2021). Traditional metrics, such as concept accuracy, fail to capture these nuances, necessitating a more comprehensive evaluation framework.

Motivated by the need for robust and interpretable evaluations, we propose a novel approach tailored to unsupervised CBMs. Our framework encompasses a suite of metrics designed to quantitatively assess concept quality across multiple dimensions, including relevance, consistency, and informativeness. We introduce ConceptScore, which leverages the Long-CLIP Zhang et al. (2024) architecture to measure the semantic coherence between predicted concepts and data points, and Ref-ConceptScore, an extension that incorporates ground-truth labels for a more comprehensive evaluation when available.

To provide a holistic perspective, we integrate established natural language processing (NLP) metrics like BLEUPapineni et al. (2002), METEORBanerjee & Lavie (2005), and ROUGELin (2004), complementing quantitative measures with qualitative assessments through GPT-based scoring and human evaluations. We validate the consistency and reliability of our proposed metrics by comput-

Figure 1: Overview of the framework.

ing Kendall $\tau$ correlation coefficients with human and GPT judgments, demonstrating strong certain alignments.

**Our contributions** are summarized as follows:

1. We propose a comprehensive evaluation framework for unsupervised CBMs, incorporating a range of metrics tailored to assess concept quality from multiple angles.

2. We introduce ConceptScore and Ref-ConceptScore, which leverage Long-CLIP and ground-truth labels, respectively, to provide more accurate and contextual evaluations.

3. We integrate NLP metrics and qualitative assessments to offer a well-rounded view of model performance, enhancing the interpretability of evaluation results.

Our team is committed to open sourcing the entire set of standards to the community and developing effective packages and apis to serve the community.

## 2 RELATED WORK

In the field of deep learning, model interpretability has remained an important yet challenging topic. Neural network models are often perceived as "black boxes," and their lack of transparency can lead to user distrust in model predictions, posing potential risks in their application Loi et al. (2022). To address this issue, researchers have been exploring methods to enhance the interpretability of deep learning models. Early work primarily focused on local interpretability, employing techniques that identify the most significant parts of the input data for specific decisions, thus providing approximate explanations for model predictions Ribeiro et al. (2016); Lundberg & Lee (2017). However, these locally approximated methods do not always guarantee accuracy and may require substantial subjective analysis. As research continues to advance, conceptual bottleneck models are gradually being integrated into mainstream interpretability efforts.

### 2.1 CONCEPT BOTTLENECK MODELS

CBMs have been extensively studied in interpretability by introducing an intermediate layer, where interpretable concepts are mapped to neurons in the middle layers. Koh et al. (2020a) introduced the CBMs , which incorporates an intermediate concept layer to guide the model's focus on high-level concepts relevant to the prediction class during the inference process. This approach improved interpretability by introducing concepts and allowed the correction of final predictions by adjusting erroneous concepts, thereby enhancing model accuracy. Espinosa Zarlenga et al. (2022) proposed the Concept Embedding Model (CEM) , which addresses the trade-off between accuracy and interpretability in existing CBMs by learning high-dimensional concept representations. CEM provides robust concept explanations while maintaining high task accuracy and supports efficient test-time concept interventions. Yuksekgonul et al. (2023) developed the Post-hoc Concept Bottleneck Model (PCBM), which can derive concept representations from textual descriptions using multimodal models without concept labels. Then, the model maps these concept subspaces to an interpretable pre-

dictor. This approach transforms pre-trained models into CBMs and introduces a residual modeling step to restore the original prediction performance, ensuring that the model's initial predictive capabilities are preserved. Oikarinen et al. (2023) proposed a Concept Bottleneck Model framework that does not require concept annotations. This framework utilizes GPT-3 to generate and filter concept sets. It introduces a sparse prediction layer to highlight the importance of relevant concepts by mapping the backbone network's features to an interpretable concept space without annotation. This further improves the model's interpretability and detection capability.

However, there has been less attention to evaluation metrics for interpretability, particularly in unsupervised concept learning, where objective and quantitative methods for assessing interpretability are notably lacking.

## 2.2 Concepts Evaluation methods

Most existing evaluation methods assess interpretability by focusing on the accuracy of the concepts Wah et al. (2011); Nevitt et al. (2006); Koh et al. (2020a) or by treating concept learning as embeddings in high-dimensional spaces, analyzing the consistency of information within these embeddings Espinosa Zarlenga et al. (2022). However, these approaches rely heavily on labeled concepts, lacking clear interpretability metrics for unsupervised concept bottleneck models.

Therefore, this paper first identifies two key issues in unsupervised explainable evaluation. The first issue is the lack of quantifiable automatic evaluation methods, and the second is the need for these methods to exhibit a high correlation with human judgments. In the context of automated unsupervised interpretability assessment, the emphasis is on evaluating the consistency between the generated concepts and the corresponding features of the images. Utilizing pre-trained cross-modal models to obtain consistent assessments of descriptive features Hessel et al. (2021) provides valuable insights. Similarly, even in scenes with concept annotations, discriminative methods only based on natural language matching fail to adequately incorporate image features, leading to evaluation challenges and difficulties in achieving high consistency with human judgmentsStefanini et al. (2023).

Based on the analysis above, our framework addresses the identified needs for explainable evaluation. It leverages cross-modal models to assess the generated concepts from multiple dimensions, incorporating semantic and visual features. This approach ensures a comprehensive consideration of the alignment between descriptions and features, achieving a high consistency level with human judgment results.

## 3 Method

### 3.1 Notation for General CBMs

Building upon the notation established by Koh et al. (2020b), we introduce CBMs. Consider a classification problem defined over a pre-defined set of concepts $\mathcal{C} = \{c^1, \ldots, c^L\}$ and a training dataset $\mathcal{D} = \{(x_i, \mathbf{c}_i, y_i)\}_{i=1}^n$. Here, for each instance, $i$ within the dataset, $x_i \in \mathbb{R}^d$ represents the feature vector, $y_i \in \mathbb{R}^{d_z}$ denotes the label vector with $d_z$ being the dimensionality corresponding to the number of classes and $\mathbf{c}_i \in \mathbb{R}^L$ signifies the concept vector, where the $k$-th entry indicates the weight or relevance of the concept $c^k$. Within the framework of CBMs, the primary goal is to learn two distinct mappings. The first mapping, denoted by $g : \mathbb{R}^d \to \mathbb{R}^L$, transforms the input feature space into the concept space. The second mapping, $f : \mathbb{R}^L \to \mathbb{R}^{d_z}$, operates on the concept space to generate predictions in the output space. For any given input $x$, the model strives to produce a predicted concept vector $\hat{\mathbf{c}} = g(x)$ and a final prediction $\hat{y} = f(g(x))$, ensuring that both $\hat{\mathbf{c}}$ and $\hat{y}$ are as close as possible to their true values $\mathbf{c}$ and $y$, respectively.

### 3.2 ConceptScore

**Model.** In this method, we utilize CLIP Radford et al. (2021), and LongCLIP Zhang et al. (2024) as encoders for aligning images with concepts. CLIP is a cross-modal retrieval model that has achieved an understanding of cross-modal data through training on a vast number of images and their corresponding descriptions. However, the description of concepts within images often involves combinations of multiple concepts, which renders CLIP's original token limit of 77 inadequate for

capturing the completeness of these expressions. Moreover, the original CLIP model struggles to capture fine-grained features within images Yamada et al. (2024), making it challenging for concepts to align with the relevant detailed characteristics. Therefore, We employ LongCLIP for concept descriptions and image encoding to address this issue. LongCLIP supports longer contexts by employing Knowledge-Preserved Stretching and Primary Component Matching strategies, enhancing input length while improving the model's ability to distinguish detailed features. This provides a robust foundation for our multi-concept evaluation framework.

**ConceptScore** is a process that assesses the alignment between an image and a given concept using the pre-trained CLIP model. The Image Encoder, denoted as $\mathcal{E}_i$, extracts features from the input image $x$, while the Text Encoder, $\mathcal{E}_t$, processes the concept description $\mathcal{P}(\hat{c})$, where $\mathcal{P}$ represents the prompt design that contextualizes the concept in a natural language sentence. The prompt is crucial for effectively communicating the concept to the model and can be formulated as $\mathcal{P}(\hat{c})$. See the Appendix C for a detailed description of the Prompt.

To quantify the alignment, we compute the cosine similarity between the image and concept embeddings, which measures the degree of correlation between the two. However, to ensure a positive and normalized score, we apply a max operation with zero, ensuring that negative similarities are clipped to zero. A weight factor $\omega$ is introduced to adjust the significance of the similarity score in the overall evaluation. The ConceptScore for a single image-concept pair can be formalized as:

$$ConceptScore(\mathcal{E}_i(x), \mathcal{E}_t(\mathcal{P}(\hat{c}))) = \omega \cdot \max(\cos \mathcal{E}_i(x), \mathcal{E}_t(\mathcal{P}(\hat{c})), 0) \tag{1}$$

When evaluating concepts across a sample set of images, the average ConceptScore is computed to represent the overall alignment of the concept within the dataset.

### 3.3 REF-CONCEPTSCORE

**Ref-ConceptScore** is an extension of the ConceptScore, which aims to provide a more reliable evaluation by incorporating ground truth concept annotations when available. In datasets with concept annotations, these labels can serve as a reference point for assessing the quality of the unsupervised concept predictions. By leveraging the annotated concepts, we can establish a soft reference standard that guides the evaluation process.

To achieve this, we introduce a new intermediate measure, denoted as $\mathcal{H}$, which is the harmonic mean. The harmonic mean is particularly useful when dealing with ratios or rates, as it gives more weight to lower values, ensuring that a single low ConceptScore does not dominate the overall evaluation.

Given an image $x$, the predicted concept $\hat{c}$, and the ground truth concept $c$, the Ref-ConceptScore is computed as follows:

$$Ref - ConceptScore(\mathcal{E}_i(x), \mathcal{E}_t(\mathcal{P}(\hat{c})), \mathcal{E}_t(\mathcal{P}(c))) \tag{2}$$
$$= \mathcal{H}(ConceptScore(\mathcal{E}_i(x), \mathcal{E}_t(\mathcal{P}(\hat{c}))), \max(\max(\cos(\mathcal{E}_t(\mathcal{P}(\hat{c})), \mathcal{E}_t(\mathcal{P}(c)), 0), 0))) \tag{3}$$

Here, the first term in the harmonic mean is the original ConceptScore between the image and the predicted concept. The second term is the maximum cosine similarity between the predicted concept and the ground truth concept, ensuring that the predicted concept is not only aligned with the image but also coherent with the annotated concept. By taking the harmonic mean of these two scores, we obtain a refined evaluation that balances the alignment of the image with the predicted concept and the consistency of the prediction with the ground truth.

In practice, the Ref-ConceptScore provides a more comprehensive assessment, especially for datasets with concept annotations, as it not only evaluates the image-concept alignment but also verifies the plausibility of the predicted concept against the known annotations. This refinement enhances the reliability of the evaluation and can guide the optimization of models for better concept understanding.

## 3.4 CONCEPT-BASED METRICS VIA NLP METRICS

These metrics are adapted from the NLP domain to assess the quality of concept predictions in a more fine-grained manner. These metrics, originally designed for evaluating machine translation and text summarization, are extended to evaluate the alignment and similarity between the predicted concepts and the ground truth annotations.

**BLEU$^c$.** $BLEU^c$ is a widely used NLP metric that computes the overlap between the predicted concept $\mathcal{P}(\hat{c})$ and the reference concept $\mathcal{P}(c)$. It is based on $n$-gram precision, with a brevity penalty to discourage shorter predictions. The $n$-gram precision is defined as:

$$P_n^c = \frac{\sum_{i=1}^{N} \min(\text{count}(\mathcal{P}(\hat{c})_i^n), \text{count}(\mathcal{P}(c)_i^n))}{\sum_{i=1}^{N} \text{count}(\mathcal{P}(\hat{c})_i^n)} \tag{4}$$

where $\mathcal{P}(\hat{c})_i^n$ and $\mathcal{P}(c)_i^n$ are the $n$-grams in the concept outputs and reference concept groundtruth, respectively, and count$(\cdot)$ denotes the number of occurrences. The brevity penalty is:

$$BP^c = \exp(1 - \frac{\text{len}(\mathcal{P}(\hat{c}))}{\text{len}(\mathcal{P}(c))}) \tag{5}$$

$$BLEU^c = BP^c \cdot \exp\left(\sum_{n=1}^{N} w_n \log P_n^c\right) \tag{6}$$

where $w_n$ are weights assigned to different $n$-grams, typically set to $1/N$.

**METEOR$^c$.** $METEOR^c$ combines precision, recall, and a harmonic mean of unigrams, with additional features such as stemming, synonymy, and word order. The unigram precision and recall are:

$$P_u^c = \frac{\text{matched\_unigrams}}{\mathcal{P}(\hat{c})\_\text{unigrams}} \tag{7}$$

$$R_u^c = \frac{\text{matched\_unigrams}}{\mathcal{P}(c)\_\text{unigrams}} \tag{8}$$

The harmonic mean is:

$$F_{\text{mean}}^c = \frac{2 \cdot P_u^c \cdot R_u^c}{P_u^c + R_u^c} \tag{9}$$

$METEOR^c$ also considers an alignment score and a penalty for unmatched words, resulting in the final score:

$$METEOR^c = F_{\text{mean}}^c \cdot (1 - \text{penalty}) \tag{10}$$

**ROUGE$^c$.** $ROUGE^c$ is primarily used for evaluating summaries, focusing on recall. It calculates the longest common subsequence (LCS) of $n$-grams between the concept outputs and reference concept ground truth. The $ROUGE^c$-$n$ recall is:

$$R_n^c = \frac{\text{LCS}(\mathcal{P}(\hat{c})^n, \mathcal{P}(c)^n)}{\max\_\mathcal{P}(c)\_\text{ngrams}(\mathcal{P}(c)^n)} \tag{11}$$

where $\mathcal{P}(\hat{c})^n$ and $\mathcal{P}(c)^n$ are the $n$-grams in the concept outputs and reference concept ground truth, respectively. $ROUGE^c$-$L$ measures the longest common subsequence of the longest $n$-grams:

$$R_L^c = \frac{\text{LCS}(\mathcal{P}(\hat{c})^{\max}, \mathcal{P}(c)^{\max})}{\max\_\mathcal{P}(c)\_\text{length}(\mathcal{P}(c))} \tag{12}$$

## 3.5 Human Score and LLM Score

To complement the automatic metrics, we introduce human and LLM scores that provide subjective assessments of concept understanding. While these scores are not incorporated into the comprehensive evaluation of the full test set due to their time-consuming nature, they serve as valuable references for validating and refining the automatic metrics.

For the human score, we devise a scoring protocol where a set of five volunteers are tasked with evaluating a subset of the data.

Each evaluation by a human rater takes approximately one minute or more, emphasizing the need for a concise scoring system. To maintain a manageable workload and ensure a representative sample, we limit the human evaluation to 100 randomly selected data points from each test set.

For the LLM Score, we employ GPT4-vision to mimic human judgment by feeding it the same prompt. The model then generates a score on the same 1-4 scale. This approach allows us to incorporate a machine's understanding of the image-concept relationship, providing an additional perspective on the evaluation. The prompt given to the GPT4 is:

> *Here's an image, which the model predicts to be pred, and the model recognizes the following pairs of features and weights: features_str; Consider the combination of features and weights as a unit. Rate how well the ensemble fits the image and rate it on a scale of 1- not the same, 2- partially the same, 3- basically the same, and 4- exactly the same. You just output the score (Only a number).*

Given the coarse-grained nature of these scores, we can reasonably expect a high level of consistency. The detailed description of the human evaluation protocol and the LLM scoring method is provided in Appendix I.

## 4 Experiments and Evaluations Benchmark

### 4.1 Baselines

**Concept Bottleneck Models.** are designed to leverage human-interpretable concepts to enhance model interpretability. They consist of two stages: first, the model predicts an intermediate set of human-specified concepts $c$, and then it uses these concepts to predict the final output $\hat{y}$. This design facilitates human interaction by allowing corrections of individual concepts, which can then influence the final prediction. CBMs are evaluated based on their ability to maintain high task accuracy while achieving high concept accuracy, indicating the model's alignment with the true concepts.

**Concept Embedding Models.** extend CBMs by learning high-dimensional embeddings for each concept. Instead of a single scalar value per concept, CEMs learn a pair of embeddings $\hat{c}_+$ and $\hat{c}_-$ for each concept, representing the active and inactive states, respectively. This design allows for richer representations and more effective interventions, as the model can switch between these states during inference. CEMs are evaluated on their ability to maintain or improve task accuracy while providing robust concept-based explanations and effective interventions.

**Label-free Concept Bottleneck Models.** are a variant that aims to overcome the need for labeled concept data, which can be time-consuming and labor-intensive to collect. They use a projection method to align neurons in a neural network with human-understandable concepts, leveraging techniques such as CLIP to create concept alignments without requiring additional labeled data. This approach is scalable, efficient, and automated.

**Post-hoc Concept Bottleneck Models.** are designed to convert any existing neural network into a CBM without the need for concept annotations during training. They achieve this by using multimodal models to infer concepts from the model's internal representations. PCBMs maintain the original model's performance while offering interpretability benefits. Additionally, PCBMs can be updated with user feedback, allowing for quick debugging and updating to reduce spurious correlations and improve generalization.

More about the baselines can be found in the Appendix A.

## 4.2 DATASETS AND SETTINGS

Our evaluation framework is designed to assess concept understanding across a diverse range of image datasets, including both labeled and unlabeled data. We choose Cifar10/100 and CUB200 to represent these categories.

**Cifar10/100.** Cifar10 and Cifar100 are widely used benchmark datasets for image classification tasks, consisting of 60,000 32x32 color images in 10 and 100 classes, respectively. These datasets are primarily unlabeled in terms of concepts, which allows us to test the generalization capabilities of our models in identifying and understanding concepts without explicit supervision. We split each dataset into training, validation, and test sets following the standard protocol.

**CUB200.** The Caltech-UCSD Birds-200-2011 (CUB200) dataset is a more specialized and challenging collection, focusing on bird species classification. It contains 200 classes with 11,788 images, each annotated with bounding boxes, part locations, and 312 binary attributes that represent various visual concepts. This labeled dataset enables us to evaluate the models' performance in recognizing and understanding specific concepts within the images. We use the provided train-test split for our experiments.

In both datasets, we preprocess the images by resizing them to a standard size and normalizing the pixel values. For CUB200, we utilize the concept annotations to compute the Ref-ConceptScore and evaluate the concept-based metrics. The human and LLM scoring is performed on a subset of images from both datasets, as described in Section 3.5. The choice of these datasets allows us to analyze the models' performance in both general and domain-specific scenarios, providing a comprehensive assessment of their concept understanding abilities.

**Settings.**

We conducted experiments using the NVIDIA GTX 3090 Ti GPU, with Python version 3.8, CUDA version 11.3, and PyTorch version 1.11.0. The training processes for all models followed the official default parameter settings. In selecting the concepts, we considered the weight and the degree of influence, analyzing the top 3, 5, 8, 12, and 15 pairs of concepts and their corresponding weight values. The selected concepts effectively encompass the features represented in the images. To account for the impact of different prompts on concept descriptions, we also designed five distinct prompt formats to enrich the validation process.

## 4.3 COMPOSITE CORRELATIONS.

In order to gain a comprehensive understanding of the relationships between the various evaluation methods, we perform a composite correlation analysis that examines the interplay between the automatic metrics, human scores, and GPT scores. This analysis aims to identify the extent to which these different measures align and complement each other, providing insights into the strengths and weaknesses of the individual evaluation components. The calculated inter-rater consistency for the five human evaluators, with a Krippendorff's alpha of 0.7405, indicates a high level of agreement among the raters, demonstrating the reliability of their scores and highlighting the robustness of the human evaluation component in assessing concept understanding.

### 4.3.1 ANALYSIS OF RESULTS

The analysis of the composite correlation matrix reveals several key findings:

**Human and GPT Correlations:** The correlation between human scores and GPT scores is 0.465658907, which serves as a proxy for the consistency of the machine-generated approximations of human judgment. This moderate correlation suggests that the GPT model provides a reasonable approximation of human perception, but not a perfect substitute.

**Human vs. Automatic Metrics:** The correlations between human scores and the automatic metrics (ConceptScore: 0.4213, Ref-ConceptScore: 0.4704, BLEU: 0.3083, METEOR: 0.3076, and ROUGE: 0.2946) indicate that the models' concept predictions align moderately well with human perception. The lower correlations suggest that the models still struggle to capture the full complexity of human understanding.

**GPT vs. Automatic Metrics:** The correlations between GPT scores and the automatic metrics (ConceptScore: 0.4299, Ref-ConceptScore: 0.5913, BLEU: 0.2506, METEOR: 0.2014, and ROUGE: 0.2145) reveal that the GPT model aligns more closely with the Ref-ConceptScore, indicating that the automatic metric capturing the consistency with ground truth concepts is more informative in terms of aligning with the GPT model's understanding.

These findings suggest that while there is a moderate level of alignment between human, GPT, and automatic metrics, there is still room for improvement in capturing the nuances of human understanding. The identified correlations can guide future improvements in model development and evaluation, ultimately contributing to more trustworthy and interpretable AI systems.

### 4.4 WITHOUT REFERENCE

**Evaluations.** The table 1 presents the evaluation scores for unsupervised data of the top 5 combinations for Prompt1. More detailed results can be found in the appendix. By calculating the ConceptScore for each sample and averaging the results, we obtain a corresponding quantitative and interpretable evaluation. On the CIFAR-10 and CIFAR-100 datasets, LFCBM achieved ConceptScores of 0.4767 and 0.4911, respectively, outperforming PHCBM by 1.44% and 3.01%. This upward trend is consistent with both human and GPT-4v evaluations, further validating the objectivity of the evaluation framework.

Table 1: Comparison of the results in cifar datasets.

| models/datasets | Prompt_Topk | cifar10 | cifar100 |
|---|---|---|---|
| phcbm | 1_5 | 0.4623 | 0.4610 |
| lfcbm | 1_5 | 0.4767 | 0.4911 |

### 4.5 WITH REFERENCE

**Evaluations.** The table 2 presents the evaluation scores for supervised data of the top 5 combinations for Prompt1. More detailed results can be found in the appendix. We evaluated the entire dataset using ConceptScore, Ref-ConceptScore, and traditional NLP metrics. Notably, the unsupervised concept labeling method, LFCBM, achieved the best performance in both ConceptScore and Ref-ConceptScore. This result aligns closely with GPT-4v and human evaluations. Although traditional NLP metrics are informative, for the task of concept evaluation, the consistency between concept descriptions and image features is more critical. The evaluation framework for this consistency demonstrates the objectivity and accuracy of our approach in assessing concepts. We also discuss the fairness of LFCBM in the evaluation phase in Appendix D.

Table 2: Comparison of the results in cub200 dataset.

| models | Prompt_Topk | ConceptScore | Ref-ConceptScore | BLUE | METEOR | ROUGE |
|---|---|---|---|---|---|---|
| cbm | 1_5 | 0.3846 | 0.5427 | 0.1198 | 0.1107 | 0.1759 |
| cem | 1_5 | 0.3851 | 0.5432 | 0.1156 | 0.1110 | 0.1694 |
| lfcbm | 1_5 | 0.3999 | 0.5563 | 0.0714 | 0.0925 | 0.1043 |
| lfcbm_unsupervise | 1_5 | 0.4533 | 0.6007 | 0.0607 | 0.0252 | 0.0667 |
| phcbm | 1_5 | 0.3951 | 0.5531 | 0.1057 | 0.1039 | 0.1618 |

### 4.6 SENSITIVITY ANALYSIS AND CASE STUDIES

We conducted experiments involving the replacement and modification of generated concepts to validate the effectiveness of our evaluation framework further. As illustrated in the figure 2, we first replaced incorrect concepts predicted by the original model, resulting in a noticeable decrease in the ConceptScore from 0.5625 to 0.3811, and the Ref-ConceptScore from 0.6958 to 0.5297. This demonstrates that our evaluation framework produces significant score differences when concepts are misaligned with the predicted image. We also tested the framework's sensitivity to the importance and weight of the concepts. By rearranging the order of the top 8 concepts and altering their corresponding weights, we observed that, despite all concepts being derived from model predictions, the ConceptScore decreased from 0.5625 to 0.5136. In contrast, the Ref-ConceptScore decreased from 0.6958 to 0.6513. These results further indicate that our framework is sensitive to the importance of concepts, enabling it to discern critical concepts within the image effectively.

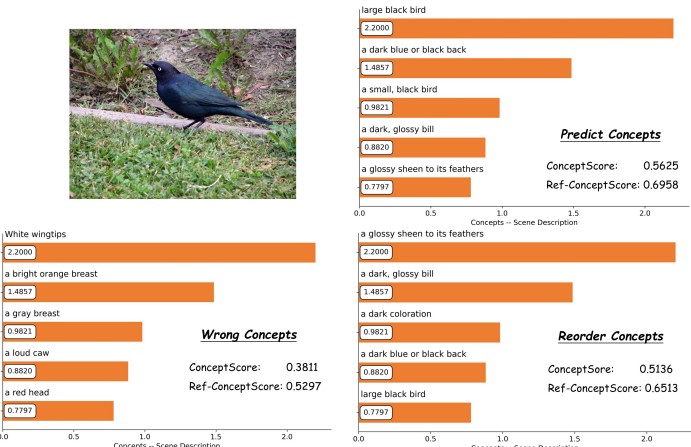

Figure 2: Concepts sensitivity. We manually modified the predicted concepts of the model and rearranged their order. In the visualized results, we adjusted the positions of the top 8 predicted concepts, and the top 5 are shown in the figure. This adjustment also involved modifying the corresponding concept weights.

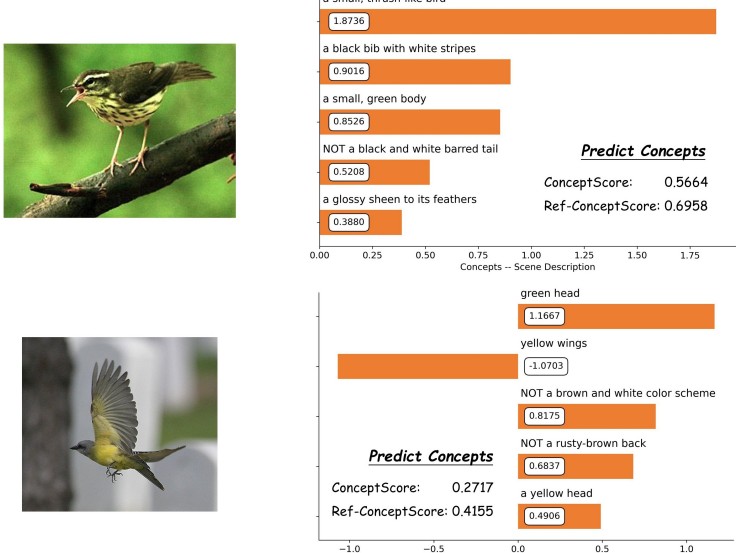

Figure 3: Concepts results.For the same model and dataset, our framework remains effective in evaluating the quality of concepts. It assigns higher scores to well-represented concepts and lower scores to those with poorer expression.

## 5 CONCLUSION

In this paper, we have proposed a novel evaluation framework for unsupervised CBMs, introducing metrics such as ConceptScore and Ref-ConceptScore to assess the quality of generated concepts without reliance on ground-truth labels. Our comprehensive approach, incorporating both quantitative and qualitative assessments, demonstrates strong correlations with human judgments and LLM evaluations, highlighting its effectiveness in enhancing the interpretability of AI systems. Through rigorous experimentation and sensitivity analyses, we have validated the framework's ability to discern critical concepts and their alignments, significantly contributing to the field of explainable artificial intelligence. Our work not only addresses a crucial challenge in evaluating unsupervised CBMs but also lays the groundwork for future research aimed at making AI more transparent and understandable.

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

## A  BASELINES

### A.1  DETAILED INTRODUCTION TO CONCEPT EMBEDDING MODELS (CEMS)

Concept Embedding Models (CEMs) extend traditional concept bottleneck models by learning high-dimensional embeddings for each concept. Instead of representing concepts as binary or scalar values, CEMs learn embeddings that represent the active and inactive states of each concept. This design facilitates the creation of models that can switch between these states during inference, thereby enabling effective human interventions. Mathematically, for each concept $c_i$, CEMs learn a pair of embeddings $\hat{c}_i^+$ and $\hat{c}_i^-$, representing the active and inactive states, respectively. These embeddings are derived from a hidden representation $h$ of the input $x$:

$$\hat{c}_i^+ = \phi_i^+(h) = a(W_i^+ h + b_i^+) \tag{13}$$

$$\hat{c}_i^- = \phi_i^-(h) = a(W_i^- h + b_i^-) \tag{14}$$

where $a(\cdot)$ is an activation function, and $W_i^+, W_i^-$ and $b_i^+, b_i^-$ are learnable parameters. The final concept embedding $\hat{c}_i$ is a convex combination of the active and inactive embeddings:

$$\hat{c}_i = p_i \hat{c}_i^+ + (1 - p_i)\hat{c}_i^- \tag{15}$$

where $p_i$ is the probability that the concept $c_i$ is active. This architecture ensures that CEMs maintain high task accuracy while providing meaningful concept representations that can be effectively manipulated at test time.

### A.2 POST-HOC CONCEPT BOTTLENECK MODELS (PCBMS)

Post-Hoc Concept Bottleneck Models (PCBMs) are designed to transform any existing neural network into an interpretable CBM without requiring concept annotations during training. PCBMs achieve this by inferring concepts from the model's internal representations using multimodal models. This process involves extracting internal representations from the trained model and mapping these representations to human-understandable concepts using multimodal models. The inferred concepts are then incorporated into the model's decision-making process, allowing for concept-level interventions.

Mathematically, PCBMs learn a mapping $g$ from the internal representations $h$ to a set of inferred concepts $\hat{c}$:

$$\hat{c} = g(h) \tag{16}$$

where $g$ is a function learned using multimodal models. The final prediction $\hat{y}$ is made using the inferred concepts:

$$\hat{y} = f(\hat{c}) \tag{17}$$

where $f$ is the downstream task predictor. This approach ensures that PCBMs maintain the original model's performance while providing interpretability benefits. By enabling concept-level feedback, PCBMs can be efficiently updated to reduce spurious correlations and improve generalization.

### A.3 LABEL-FREE CONCEPT BOTTLENECK MODELS (LFCBMS)

Label-Free Concept Bottleneck Models (LFCBMs) are designed to enhance the interpretability of neural networks without the need for labeled concept data. This approach automates the process of generating concept bottleneck models, making it scalable and efficient for large datasets. Initially, a set of concepts is generated using generative models like GPT-3, tailored to the classes in the dataset. This set is then filtered to remove concepts that are too long, too similar to the output classes, or redundant with each other. The remaining concepts serve as the basis for the bottleneck layer. Embeddings for both the backbone network's output and the concept set are computed using a multimodal model like CLIP. Projection weights $W_c$ are then learned to map the backbone network's activations to the concept embeddings, ensuring alignment between the model's internal representations and the human-interpretable concepts.

Finally, a sparse final layer $W_F$ is trained to make predictions based on the projected concepts. This sparsity ensures that the model's decisions are influenced by a few key concepts, enhancing interpretability. Mathematically, the LFCBM's prediction process can be described as $\hat{c} = \sigma(W_c \cdot \text{Backbone}(x) + b)$ and $\hat{y} = W_F \cdot \hat{c}$, where $\sigma$ is an activation function (typically sigmoid), $W_c$ are the projection weights, $b$ is a bias term, $\hat{c}$ is the projected concept vector, and $\hat{y}$ is the predicted label. This streamlined architecture ensures that LFCBMs are both interpretable and maintain high accuracy, making them a practical solution for deploying interpretable models in real-world applications.

## B THEORETICAL FOUNDATIONS OF CONCEPT-BASED METRICS

### B.1 CONCEPT REPRESENTATION AND ALIGNMENT

The concept representation in CBMs is rooted in the idea of decomposing complex visual information into a set of interpretable and meaningful components. This decomposition allows for a more transparent understanding of the model's decision-making process and enables the evaluation of individual concepts. The alignment between image features and concept representations is crucial for effective concept-based evaluation. **Cross-Modal Alignment.** The use of CLIP and LongCLIP in the ConceptScore and Ref-ConceptScore is based on the concept of cross-modal alignment Radford et al. (2021); Zhang et al. (2024). These models learn to align image and text representations in a shared embedding space, enabling the comparison of visual and textual information. The cosine

similarity between image and concept embeddings quantifies the degree of alignment, reflecting the model's understanding of the concept within the image.

## B.2 HARMONIC MEAN AND WEIGHTED SUM

The harmonic mean in the Ref-ConceptScore is a principled choice for combining the alignment and consistency scores. It is a weighted average that gives more weight to smaller values, ensuring that a single low score does not dominate the overall evaluation. The weight factor $\omega$ in the ConceptScore allows for adjusting the relative importance of the alignment score, providing flexibility in the evaluation. **Theoretical Properties.** The harmonic mean has the following properties that make it suitable for our purpose:

1. **Monotonicity:** If $a \leq b$, then $\mathcal{H}(a, b) \geq \mathcal{H}(a', b')$ if $a' \leq a$ and $b' \leq b$.
2. **Boundness:** $\mathcal{H}(a, b) \leq \min(a, b)$, ensuring that the combined score is bounded by the minimum of the individual scores.
3. **Weighted Average:** The harmonic mean can be seen as a weighted average with weights inversely proportional to the values, providing a balance between high and low scores.

These properties ensure that the Ref-ConceptScore provides a balanced and robust evaluation of concept understanding.

## B.3 NLP METRICS FOR CONCEPT EVALUATION

The adaptation of NLP metrics, such as BLEU, METEOR, and ROUGE, for concept-based evaluation relies on the analogy between text sequences and concept representations. These metrics are designed to measure the similarity between sequences, which can be extended to evaluate the similarity between predicted and ground truth concepts. **Overlap and Diversity.** The $n$-gram precision, recall, and longest common subsequence used in these metrics capture the overlap between the predicted and reference sequences while considering the diversity of the concepts. This approach ensures that the evaluation accounts for both the presence and the order of concepts. **Stemming and Synonymy.** Incorporating stemming and synonymy in METEOR and ROUGE acknowledges the variations in word forms and the semantic equivalence of concepts, enhancing the evaluation's robustness. **Penalties and Length Normalization.** Penalties and length normalization in these metrics ensure that shorter or less diverse predictions are not favored over longer or more comprehensive ones, providing a more comprehensive assessment of concept understanding.

The theoretical foundations of our concept-based evaluation framework are rooted in cross-modal alignment, weighted averaging, and sequence similarity measures, ensuring a well-justified and rigorous assessment of concept understanding in CBMs.

## C PROMPT AND THRESHOLD VALUE FOR SCORE

To thoroughly account for the impact of different prompts on the evaluation framework, we paired relevant concepts with corresponding weights based on varying linguistic preferences. We input them into the evaluation framework for assessment. The various forms of the prompts are shown in the table 3.

The evaluation of the five distinct prompts for ConceptScore revealed a remarkable consistency in their correlation with human scores. The correlation coefficients, which were 0.4027, 0.4201, 0.4213, 0.3917, 0.4088 for each prompt, indicate a minimal influence of the prompt choice on the overall concept understanding assessment. Despite this observation, it is noteworthy that the subtle differences in prompt phrasing did not significantly impact the alignment between the model's concept predictions and human perception.

In the main experimental section, we opted for the prompt that exhibited the highest correlation with human scores, as it provides the most reliable representation of concept understanding. This choice underscores the importance of selecting an optimal prompt for ensuring a robust evaluation. However, the near-identical correlations across all prompts suggest that the ConceptScore is relatively insensitive to prompt variations, offering a stable metric for concept evaluation.

In the Appendix, we present a comprehensive analysis of the results from all five prompts, detailing the correlation coefficients and discussing any potential factors that might have contributed to the observed consistency. This comprehensive report provides a more complete picture of the prompt sensitivity in the ConceptScore and serves as a valuable reference for future studies exploring the impact of prompt design on concept-based evaluations.

Table 3: Prompt Variants

**Prompt Variants**

1. Please enter your features (concepts) and their corresponding weights in the following format:
concept1: weight1, concept2: weight2, concept3: weight3...

2. The features (concepts) and weights in a table format:

| Concept | Weight |
|---------|--------|
| concept1 | weight1 |
| concept2 | weight2 |
| concept3 | weight3 |
| ... | ... |

3. The image is influenced by the following features (concepts) and their associated weights:
concept1: weight1, concept2: weight2, concept3: weight3...
Rank the importance based on the weight's absolute value.

4. Paired Features: Each feature (concept) is paired with its weight to indicate its relevance to the image:
concept1: weight1, concept2: weight2, concept3: weight3...
Rank the importance based on the weight's absolute value.

5. The image's interpretation is shaped by the following features (concepts), ranked by their weight significance:
concept1: weight1, concept2: weight2, concept3: weight3...
Rank the importance based on the weight's absolute value.

## D  FAIRNESS OF LFCBM

To validate the impact of different expressions of concepts on LFCBM, we compared the concept expressions used in CBMs with the self-generated concepts defined by LFCBM. The evaluation results are summarized in the table 4. In terms of traditional NLP metrics, the concept annotations in the cub200 dataset follow the CBMs format, leading to higher NLP metrics when using the CBMs concept expression format. However, regarding the alignment between concepts and image features, self-generated concepts, which better align with the characteristics of feature representation, achieve a higher ConceptScore, and even the Ref-ConceptScore shows a corresponding improvement. The comparative results of the experiments further illustrate that this evaluation framework emphasizes relevant consistency at the conceptual description and image feature levels, rather than traditional annotation prediction subword matching, thus better meeting the evaluation requirements for concepts.

Table 4: Fairness of LFCBM.

| concepts | ConceptScore | Ref-ConceptScore | BLUE | METEOR | ROUGE |
|----------|--------------|------------------|------|--------|-------|
| CBMs concepts | 0.3999 | 0.5563 | 0.0714 | 0.0925 | 0.1043 |
| Unsurvised | 0.4533 | 0.6007 | 0.0607 | 0.0252 | 0.0667 |

## E  COMPUTATION COST

To validate the efficiency of our evaluation framework, we conducted an average evaluation time analysis for each method. As shown in the table 5, our approach demonstrates significant efficiency across various datasets. When using GPT-4v for evaluation, the average time per sample was 20 seconds, with additional communication overhead considered. For human evaluation, despite using convenient tools like a UI scoring interface, the average time per sample ranged from 120 to 180 seconds (Note that this is just the coarse-grained rating). In contrast, when using CLIP for evaluation, our framework achieved an average evaluation time of only 42.24 seconds on the CUB-200 dataset, which contains 5,794 images with annotated concepts. The average time for the unsupervised CIFAR-10 and CIFAR-100 datasets was merely 9.57 and 8.84 seconds, respectively. Even when using Long-CLIP as the baseline model, the evaluation time cost was only 0.065% of that required by GPT-4v, and 0.01% of that of human evaluation, highlighting the efficiency and low-cost nature of our approach.

Table 5: Time cost on datasets.

| datasets | image counts | clip time(s) | long-clip times(s) | GPT time(s) | Human time(s) |
|---|---|---|---|---|---|
| cub200 | 5794 | 42.24 | 75.63 | 20 × 5794 | 130 × 5794 |
| cifar10 | 10000 | 9.57 | 57.12 | 20 × 10000 | 120 × 10000 |
| cifar100 | 10000 | 8.84 | 57.47 | 20 × 10000 | 120 × 10000 |

## F  MORE CASE STUDIES AND SENSITIVITY EXPERIMENTS

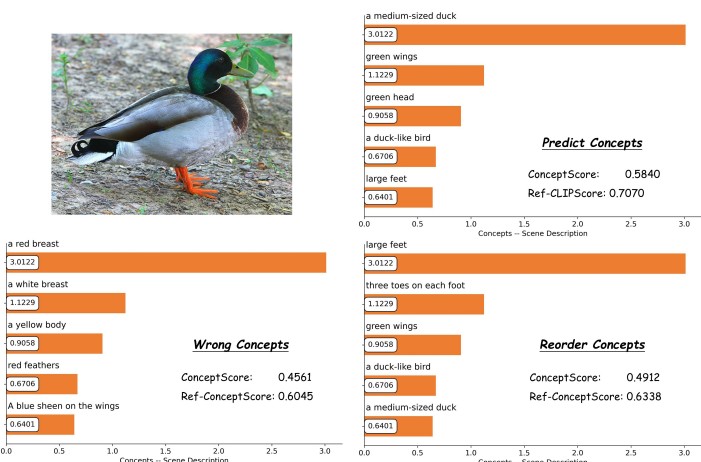

Figure 4: Concepts sensitivity case on cub200 dataset.

## G  DETAILS OF KENDALL $\tau$

Kendall's rank correlation coefficient, denoted as $\tau$, is a non-parametric measure of the strength and direction of the relationship between two variables. It is particularly useful when assessing the association between ordinal variables, such as the scores assigned by different evaluation methods. In our case, we use $\tau$ to quantify the correlation between the automatic metrics, human scores, and GPT scores.

The $\tau$ coefficient is based on the number of concordant and discordant pairs in the data. Concordant pairs are those where both variables increase or decrease together, while discordant pairs are those where one variable increases as the other decreases. The formula for Kendall's $\tau$ is given by:

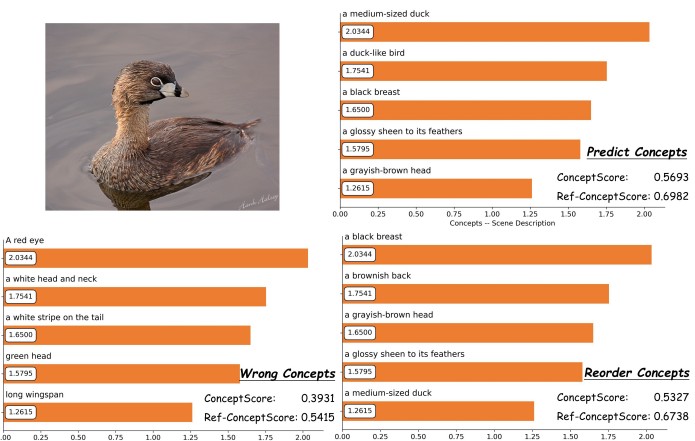

Figure 5: Concepts sensitivity case on cub200 dataset.

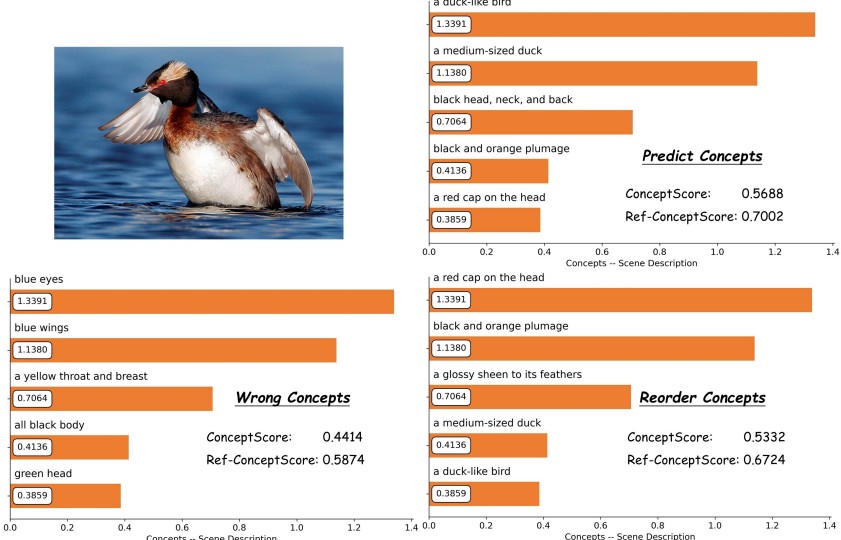

Figure 6: Concepts sensitivity case on cub200 dataset.

$$\tau = \frac{2 \times (C - D)}{n(n-1)} \tag{18}$$

where $C$ is the number of concordant pairs, $D$ is the number of discordant pairs, and $n$ is the total number of pairs.

Kendall's $\tau$ ranges from -1 to 1, with -1 indicating a perfect negative correlation, 0 indicating no correlation, and 1 indicating a perfect positive correlation. A positive $\tau$ value indicates that the variables tend to move in the same direction, while a negative value suggests they move in opposite directions.

A high positive $\tau$ between the automatic metrics and human or GPT scores would indicate a strong agreement between these methods, suggesting that the automatic metrics are effectively capturing the concept understanding. Conversely, a low or negative $\tau$ would imply that the automatic metrics are not well-aligned with human or GPT evaluations, requiring further refinement.

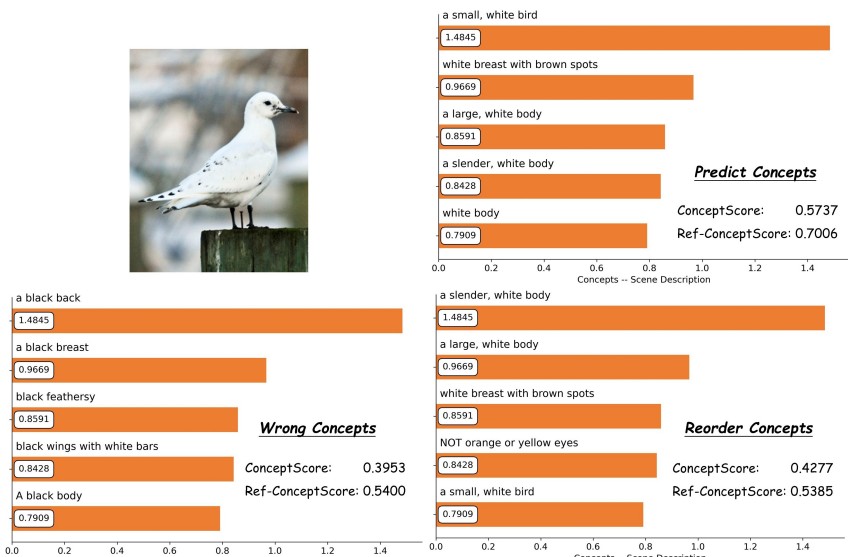

Figure 7: Concepts sensitivity case on cub200 dataset.

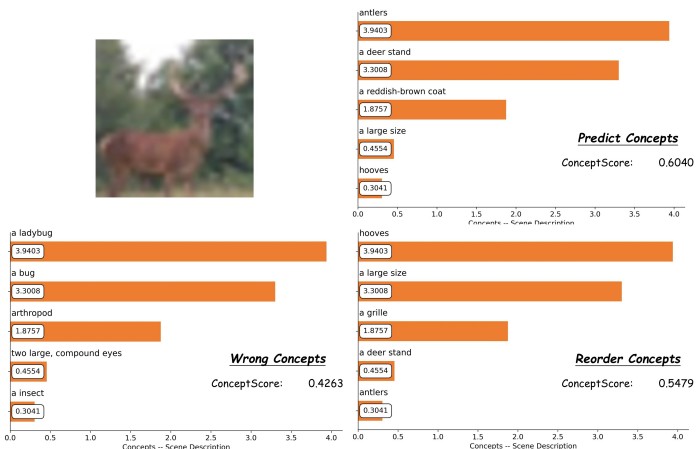

Figure 8: Concepts sensitivity case on cifar10 dataset.

# H WHY ESTABLISH THE CORRELATION OF THE THREE TYPES OF RATINGS?

Our evaluation framework should aim to establish a correlation between all three types of scores: automatic metrics, human scores, and GPT scores. The reason for this is to ensure a well-rounded understanding of the model's performance and concept understanding.

1. **Automatic Metrics:** These metrics provide a fast and scalable way to assess the model's performance, but they may not fully capture the nuances of human perception.

2. **Human Scores:** Human scores offer a subjective and context-aware evaluation, reflecting the complexity of human understanding. However, they are time-consuming and may not be feasible for large-scale evaluations.

3. **GPT Scores:** GPT scores serve as an approximation of human judgment, providing a middle ground between automatic metrics and human scores. They are faster than human evaluations but may not fully align with human perception.

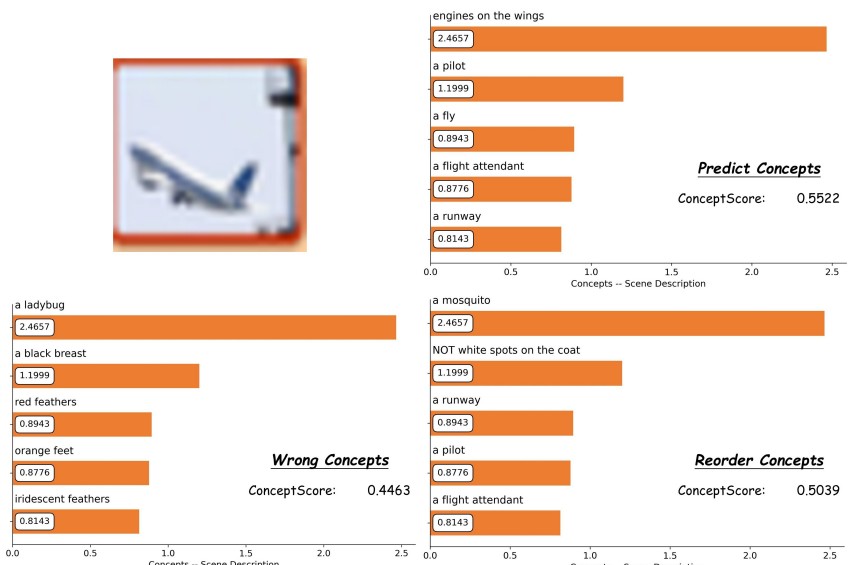

Figure 9: Concepts sensitivity case on cifar10 dataset.

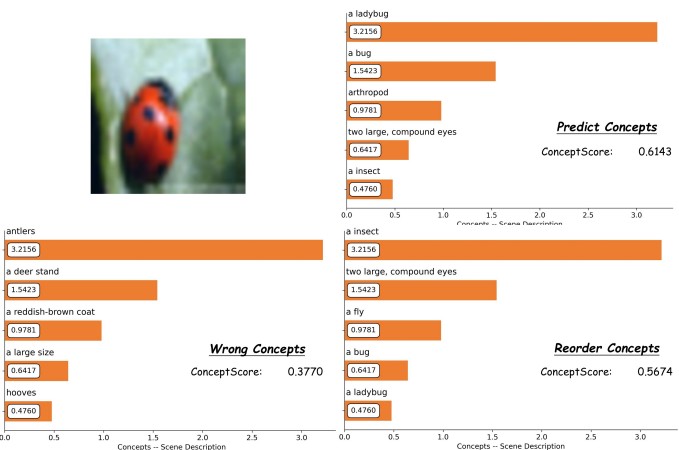

Figure 10: Concepts sensitivity case on cifar100 dataset.

By computing the correlation between all three types of scores, you can:

1. Validate the automatic metrics by comparing them with human and GPT scores. A strong correlation would indicate that the automatic metrics are capturing relevant aspects of concept understanding.

2. Identify the strengths and limitations of the GPT model in mimicking human judgment. A high correlation with human scores would suggest that GPT is a reliable proxy for human evaluation.

3. Refine the automatic metrics by identifying the most informative features or combinations that align well with both human and GPT scores. This can help optimize the evaluation framework and improve its predictive power.

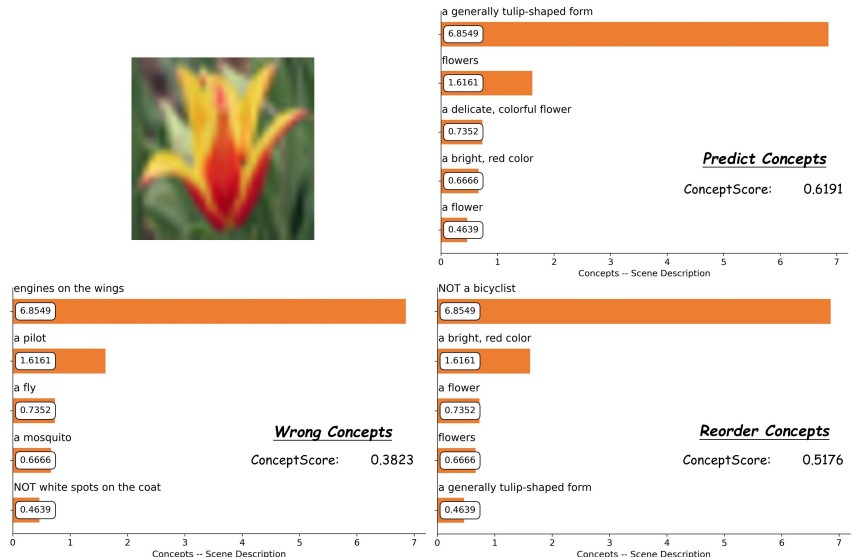

Figure 11: Concepts sensitivity case on cifar100 dataset.

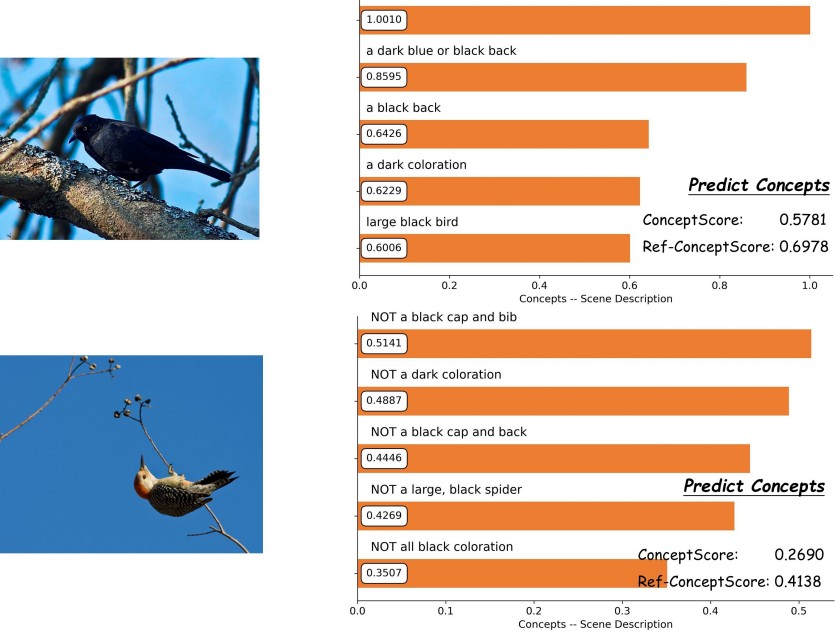

Figure 12: Concepts case on cub200 dataset.

# I   DETAILS OF HUMAN RATINGS AND LLM RATINGS

## I.1   HUMAN EVALUATION PROTOCOL

The human evaluation protocol is designed to assess the quality of the model's concept predictions in a subjective manner. The following steps outline the process:

1. **Sample Selection:** A random subset of 100 data points is selected from the test set to be evaluated by human raters.

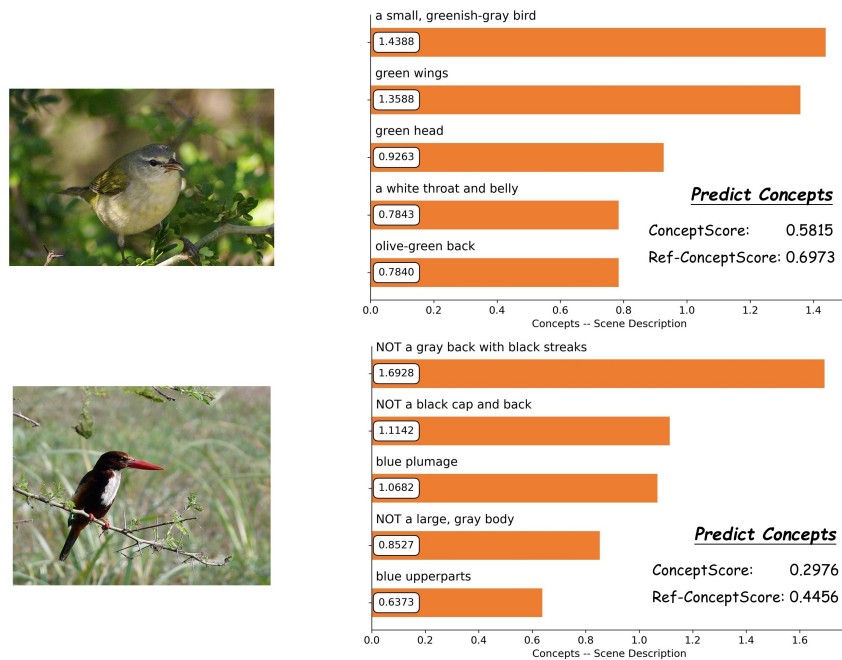

Figure 13: Concepts case on cub200 dataset.

2. **Rater Preparation:** Five volunteers, with no prior knowledge of the model or its predictions, are recruited for the task. They are provided with clear instructions on the scoring scale (1-4) and the prompt mentioned in Section 3.5.

3. **Evaluation:** Each rater independently evaluates the selected data points, assigning a score based on the prompt. The evaluation is performed in a controlled environment to minimize external influences.

4. **Consensus and Discrepancies:** The scores from the five raters are collected, and any discrepancies are resolved through discussion and majority voting, if necessary. The final human score for each data point is the average of the adjusted scores.

5. **Quality Control:** To ensure the reliability of the human scores, inter-rater reliability is calculated using the Fleiss' Kappa coefficient Kraemer (1980), which measures the agreement among the raters beyond chance.

### I.2 FLEISS' KAPPA COEFFICIENT AND INTER-RATER RELIABILITY

Fleiss' Kappa coefficient Kraemer (1980) is a statistical measure used to assess the agreement among multiple raters when assigning categorical ratings. It takes into account both chance agreement and actual agreement, providing a more robust evaluation of inter-rater reliability than simple percentage agreement. Fleiss' Kappa ranges from -1 to 1, where 1 indicates perfect agreement, 0 indicates agreement no better than chance, and negative values indicate agreement worse than chance.

In our human evaluation protocol, Fleiss' Kappa is employed to quantify the level of agreement among the five human raters. The following steps outline its application:

1. **Rating Data Collection:** The scores assigned by each rater to the 100 data points are collected.

2. **Chance Agreement Calculation:** The expected agreement by chance is computed using the marginal totals of the ratings, which represents the probability of raters agreeing by chance alone.

3. **Observed Agreement Calculation:** The actual agreement among the raters is calculated, which is the proportion of times they assign the same score to a data point.

4. **Fleiss' Kappa Computation:** Fleiss' Kappa is calculated using the observed agreement and chance agreement, as per the formula:

$$\kappa = \frac{\bar{P} - P_e}{1 - P_e} \qquad (19)$$

where $\bar{P}$ is the observed agreement and $P_e$ is the chance agreement.

5. **Interpretation:** The calculated Fleiss' Kappa value is interpreted to determine the level of agreement among the raters. A value of 0.40 or higher is generally considered as substantial agreement, while a value of 0.60 or higher indicates almost perfect agreement.

By applying Fleiss' Kappa to our human scoring, we ensure that the scores are reliable and representative of the raters' consensus, providing a strong foundation for comparing them with the automatic metrics and LLM scores.

### I.3 POST-EVALUATION INTERVIEWS AND INSIGHTS

Following the human evaluation, we conducted in-depth interviews with the five volunteers to gain a deeper understanding of their scoring criteria, employed strategies, and any noteworthy observations. These interviews provided valuable qualitative insights that complement the quantitative results.

#### I.3.1 SCORING CRITERIA AND STRATEGIES

The volunteers revealed that their scoring decisions were primarily based on the perceived relevance and coherence of the predicted concepts with the image content. The following strategies emerged from their responses:

1. **Feature-Weight Analysis:** Raters meticulously evaluated the feature-weight pairs, considering the combination of features and their relative importance in determining the overall concept.

2. **Contextual Understanding:** Volunteers attempted to interpret the image within its broader context, incorporating their prior knowledge and understanding of the scene.

3. **Comparative Evaluation:** Some raters compared the predicted concepts across multiple images to establish a consistent scoring scale.

#### I.3.2 INTERESTING OBSERVATIONS

The interviews revealed several intriguing findings:

1. **Model Performance Trends:** Raters noticed that certain models consistently outperformed others on specific types of images or concepts, highlighting the strengths and weaknesses of the models.

2. **Data Set Characteristics:** The quality and complexity of the images in the dataset influenced the difficulty of assigning scores, with more ambiguous or cluttered images leading to higher variability in ratings.

3. **Unusual Concept Combinations:** Some models produced unique or unexpected concept combinations, which sometimes aligned well with the image and other times did not, indicating the challenges in capturing the complexity of visual concepts.

4. **Human-Model Discrepancies:** Raters found that their interpretation of the image often differed from the model's predictions, emphasizing the need for better alignment between human perception and machine understanding.

**Interview Transcript:**

- **Volunteer ID:** [0]

- **Scoring Criteria:** I think the key thing that determines what this image is classified into is 1 point if it's wrong, 3 or 4 points for the concepts that have the highest contribution in my mind, can't recognize the difference between it and other kinds of images but if it's right, 2 points.

- **Interesting Observations:** One model in particular liked to output the concepts that started with not. Without any information, all the models couldn't tell the difference between green and red, they would output a random color instead. These models generally perform poorly on the cifar10 dataset, and the concept of output is generally the same for images of the same class. Sometimes the granularity of the concept is almost the same as the classification. For example, the largest concept contributing to the classification of a bed is the bed itself.

- **Volunteer ID:** [1]

- **Scoring Criteria:** Because there are many descriptions, I generally give 4 points if all of them are correct, and 3 points if there are 1-2 inaccurate ones, such as inaccurate colors, etc. 2 points is greater than 2, but the basic content can still be described, such as a xxx type of bird. In my impression, I should not give 1 point, if there is, it is completely wrong.

- **Interesting Observations:** What might be interesting is that the output of most models actually has a baseline description, there are not many examples that are completely wrong, there are some examples such as complex background, color will have some description error.

- **Volunteer ID:** [2]

- **Scoring Criteria:** For the bird dataset, since there are no color concepts such as green and red, I prefer to give high scores to birds that exhibit black, white, and brown features. In the CIFAR dataset, I tend to give high scores to the categories horse, flower, motorcycle, bicycle, and car because their conceptual features are more relevant compared to the other categories.

- **Interesting Observations:** One interesting thing I noticed was the appearance of adjectives like "love" and "mean" in the concept of the CIFAR dataset. These are usually human-assigned semantic features rather than visual features of the category. At the same time, I also found that PHCBM performs very poorly on CIFAR dataset, the concepts are the same for almost every sample, hence my scores are mostly 1 or 2. In contrast, for the bird dataset, the generated concepts are basically the same under the performance of each model, but the CIFAR dataset performs differently under different models. Some models generate a few simple concepts, while others generate more than a dozen. In addition, I observed that the bird dataset only describes black eye color, but in reality birds also have red eye color, yellow eye color, etc.

- **Volunteer ID:** [3]

- **Scoring Criteria:** I will first pay attention to whether the most important feature (the feature at first glance) is present in the given feature, for example, a black bird has a yellow top, then this feature is a more critical feature for identifying this bird. I will pay attention to how well this feature is described (if not I will give a maximum score of 3). Then I see if the feature matches the image from high to low similarity and decide what score it should be.

- **Interesting Observations:** I find that certain concepts appear more frequently, such as bill_length, which gives high similarity scores. On the contrary, some concepts appear less frequently. Does this limit the diversity of concepts?

- **Volunteer ID:** [4]

- **Scoring Criteria:** When evaluating the results of the concepts model, my scoring criteria are based on the following elements: First, I refer to the salient features in the image, such as color, shape, texture, etc., to ensure that the selected concepts are highly related to the image content. Second, I will consider the weight of each concept and give preference to those descriptions with high weight that are consistent with the image features. In addition, I evaluate the diversity and concreteness of the concepts to ensure that they accurately

capture the different details in the image. In the end, I tend to choose the first eight concepts because subsequent concepts tend to have low credibility and may interfere with the decision.

- **Interesting Observations:** While conducting the review, I noticed that some of the concepts generated by the model were very descriptive and accurately captured the unique characteristics of birds. For example, some descriptions such as "a red breast" and "dark blue or black back" demonstrate strong visual impressions, while others such as "small, black bird" may be too broad and lack specificity. In addition, I found that the frequency of certain features across different images had a significant impact on concept generation, such as a particular color being more common in a particular category, causing the model to bias towards generating relevant descriptions. This phenomenon reminded me that the diversity and representativity of a dataset is critical to model performance, and the consistency of a model in the face of different backgrounds or features is a key evaluation point.

### I.4 LLM SCORING METHOD

The LLM scoring method employs GPT4-vision to mimic human judgment. The following steps describe the process:

1. **Prompt Construction:** The same prompt used in the human evaluation is prepared for the LLM, containing the image prediction and the feature-weight pairs.

2. **Model Inference:** The prompt is fed to GPT4-vision, which generates a score on the 1-4 scale based on its understanding of the image-concept relationship.

3. **Score Calibration:** To ensure consistency with the human scores, the LLM's output is calibrated using a small validation set with known human scores. This calibration adjusts the LLM's score distribution to align with the human scoring scale.

4. **Final LLM Score:** The calibrated LLM score is assigned to each data point in the subset of 100 samples.

The human and LLM scores serve as a benchmark for evaluating the performance and validity of the automatic metrics. By comparing these scores with the automatic metrics, we can identify the most informative and reliable combinations using Shapley values and a greedy algorithm, as described in the main text.

### ETHICS STATEMENT

This research adheres to the principles of ethical research and respects the privacy and intellectual property rights of all involved parties. The datasets used in this study are publicly available, and no personally identifiable information was collected or used during the experiments. The human evaluation process was conducted with informed consent from the volunteers, who were informed about the purpose of the study and their right to withdraw at any time. The study's findings are reported transparently, and no harm or deception was inflicted upon the participants or the models evaluated.

### REPRODUCIBILITY STATEMENT

To ensure the reproducibility and transparency of our research, we provide the following details:

1. **Code and Data Availability:** The code for the proposed evaluation framework, including the implementation of the metrics and the greedy algorithm, will be made publicly available upon publication under an open-source license. The datasets used in this study, Cifar10/100 and CUB200, are publicly accessible, and the links to the datasets will be provided in the code repository.

2. **Preprocessing and Hyperparameters:** The preprocessing steps and hyperparameters used for the models and evaluation metrics are documented in the code repository. The

specific settings for the GPT4-vision model and the concept-based metrics are also included.

3. **Evaluation Protocol:** The detailed evaluation protocol for the human and GPT scoring, including the prompt design and the instructions given to the evaluators, is provided in the supplementary material.

4. **Computational Resources:** The computational resources required for running the experiments, such as hardware specifications and estimated computational time, are documented in the code repository.

5. **Random Seeds and Reproducibility:** To ensure reproducibility, we use fixed random seeds for all experiments, which will be specified in the code. This guarantees that the results can be consistently reproduced by other researchers.

6. **Documentation and Instructions:** Comprehensive documentation and instructions for running the code, reproducing the experiments, and interpreting the results will be included in the code repository.

Our team is committed to open sourcing the entire set of standards to the community and developing effective packages and apis to serve the community.

Table 6: Results on cbm in cub200 dataset.

| models | Prompt_Topk | ConceptScore | Ref-ConceptScore | BLUE | METEOR | ROUGE |
|--------|-------------|--------------|------------------|------|--------|-------|
| cbm | 1_3 | 0.3747 | 0.5252 | 0.0644 | 0.0704 | 0.1392 |
| cbm | 1_5 | 0.3846 | 0.5427 | 0.1198 | 0.1107 | 0.1759 |
| cbm | 1_8 | 0.4109 | 0.5718 | 0.2136 | 0.1678 | 0.2137 |
| cbm | 1_12 | 0.4162 | 0.5782 | 0.3316 | 0.2390 | 0.2435 |
| cbm | 1_15 | 0.4265 | 0.5877 | 0.4072 | 0.2865 | 0.2588 |
| cbm | 2_3 | 0.3622 | 0.5124 | 0.0712 | 0.0686 | 0.1387 |
| cbm | 2_5 | 0.3771 | 0.5316 | 0.1346 | 0.1086 | 0.1714 |
| cbm | 2_8 | 0.3920 | 0.5523 | 0.2327 | 0.1649 | 0.2043 |
| cbm | 2_12 | 0.3987 | 0.5598 | 0.3413 | 0.2345 | 0.2286 |
| cbm | 2_15 | 0.4027 | 0.5642 | 0.3998 | 0.2808 | 0.2404 |
| cbm | 3_3 | 0.3986 | 0.5568 | 0.1119 | 0.0756 | 0.1342 |
| cbm | 3_5 | 0.4177 | 0.5781 | 0.1716 | 0.1154 | 0.1655 |
| cbm | 3_8 | 0.4271 | 0.5889 | 0.2605 | 0.1713 | 0.2006 |
| cbm | 3_12 | 0.4331 | 0.5957 | 0.3637 | 0.2398 | 0.2294 |
| cbm | 3_15 | 0.4370 | 0.5998 | 0.4259 | 0.2845 | 0.2445 |
| cbm | 4_3 | 0.3924 | 0.5494 | 0.1083 | 0.0710 | 0.1274 |
| cbm | 4_5 | 0.4063 | 0.5655 | 0.1662 | 0.1113 | 0.1618 |
| cbm | 4_8 | 0.4227 | 0.5836 | 0.2521 | 0.1683 | 0.1978 |
| cbm | 4_12 | 0.4275 | 0.5893 | 0.352 | 0.2394 | 0.2267 |
| cbm | 4_15 | 0.4278 | 0.5896 | 0.4118 | 0.2869 | 0.2418 |
| cbm | 5_3 | 0.3964 | 0.5551 | 0.1200 | 0.0740 | 0.1314 |
| cbm | 5_5 | 0.4251 | 0.5858 | 0.1788 | 0.1144 | 0.1622 |
| cbm | 5_8 | 0.4305 | 0.5923 | 0.2648 | 0.1713 | 0.1968 |
| cbm | 5_12 | 0.4368 | 0.5995 | 0.3636 | 0.2417 | 0.2255 |
| cbm | 5_15 | 0.4426 | 0.6054 | 0.4220 | 0.2884 | 0.2405 |

Table 7: Results on cbm in cub200 dataset.

| models | Prompt_Topk | ConceptScore | Ref-ConceptScore | BLUE | METEOR | ROUGE |
|---|---|---|---|---|---|---|
| cem | 1_3 | 0.3762 | 0.5268 | 0.0610 | 0.0704 | 0.1333 |
| cem | 1_5 | 0.3851 | 0.5432 | 0.1156 | 0.1110 | 0.1694 |
| cem | 1_8 | 0.4105 | 0.5714 | 0.2092 | 0.1690 | 0.208 |
| cem | 1_12 | 0.4149 | 0.5768 | 0.3285 | 0.2418 | 0.2400 |
| cem | 1_15 | 0.4254 | 0.5863 | 0.4030 | 0.2923 | 0.2547 |
| cem | 2_3 | 0.3639 | 0.5140 | 0.0678 | 0.0686 | 0.1329 |
| cem | 2_5 | 0.3768 | 0.5311 | 0.1305 | 0.1089 | 0.1651 |
| cem | 2_8 | 0.3911 | 0.5511 | 0.2289 | 0.1661 | 0.1988 |
| cem | 2_12 | 0.3970 | 0.5578 | 0.3387 | 0.2373 | 0.2249 |
| cem | 2_15 | 0.4009 | 0.5621 | 0.3955 | 0.2864 | 0.2359 |
| cem | 3_3 | 0.3995 | 0.5578 | 0.1084 | 0.0756 | 0.1287 |
| cem | 3_5 | 0.4179 | 0.5783 | 0.1677 | 0.1157 | 0.1595 |
| cem | 3_8 | 0.4264 | 0.5881 | 0.2572 | 0.1725 | 0.1953 |
| cem | 3_12 | 0.4319 | 0.5945 | 0.3618 | 0.2422 | 0.2262 |
| cem | 3_15 | 0.4359 | 0.5985 | 0.4231 | 0.2894 | 0.2410 |
| cem | 4_3 | 0.3939 | 0.5509 | 0.1049 | 0.0709 | 0.1220 |
| cem | 4_5 | 0.4069 | 0.5660 | 0.1625 | 0.1116 | 0.1557 |
| cem | 4_8 | 0.4225 | 0.5832 | 0.2492 | 0.1696 | 0.1925 |
| cem | 4_12 | 0.4266 | 0.5883 | 0.3505 | 0.2422 | 0.2236 |
| cem | 4_15 | 0.4261 | 0.5877 | 0.4096 | 0.2926 | 0.2384 |
| cem | 5_3 | 0.3967 | 0.5554 | 0.1166 | 0.0740 | 0.1260 |
| cem | 5_5 | 0.4244 | 0.5852 | 0.1751 | 0.1147 | 0.1563 |
| cem | 5_8 | 0.4290 | 0.5908 | 0.2619 | 0.1725 | 0.1917 |
| cem | 5_12 | 0.4349 | 0.5976 | 0.3621 | 0.2445 | 0.2224 |
| cem | 5_15 | 0.4409 | 0.6037 | 0.4197 | 0.2941 | 0.2371 |

Table 8: Results on lfcbm in cub200 dataset with CBMs concepts.

| models | Prompt_Topk | ConceptScore | Ref-ConceptScore | BLUE | METEOR | ROUGE |
|---|---|---|---|---|---|---|
| lfcbm | 1_3 | 0.3955 | 0.5436 | 0.0385 | 0.0600 | 0.0869 |
| lfcbm | 1_5 | 0.3999 | 0.5563 | 0.0714 | 0.0925 | 0.1043 |
| lfcbm | 1_8 | 0.4235 | 0.5824 | 0.1241 | 0.1367 | 0.1216 |
| lfcbm | 1_12 | 0.4285 | 0.5880 | 0.1845 | 0.1876 | 0.1352 |
| lfcbm | 1_15 | 0.4388 | 0.5964 | 0.2156 | 0.2182 | 0.1408 |
| lfcbm | 2_3 | 0.3853 | 0.5345 | 0.0438 | 0.0577 | 0.0898 |
| lfcbm | 2_5 | 0.3952 | 0.5491 | 0.0821 | 0.0901 | 0.1037 |
| lfcbm | 2_8 | 0.4062 | 0.5648 | 0.1373 | 0.1336 | 0.1173 |
| lfcbm | 2_12 | 0.4097 | 0.5687 | 0.1916 | 0.1833 | 0.1273 |
| lfcbm | 2_15 | 0.4120 | 0.5708 | 0.2128 | 0.2128 | 0.1310 |
| lfcbm | 3_3 | 0.4150 | 0.5717 | 0.0795 | 0.0650 | 0.0931 |
| lfcbm | 3_5 | 0.4340 | 0.5923 | 0.1151 | 0.0970 | 0.1046 |
| lfcbm | 3_8 | 0.4406 | 0.5997 | 0.1640 | 0.1406 | 0.1178 |
| lfcbm | 3_12 | 0.4465 | 0.6060 | 0.2149 | 0.1907 | 0.1294 |
| lfcbm | 3_15 | 0.4544 | 0.6128 | 0.2381 | 0.2208 | 0.1346 |
| lfcbm | 4_3 | 0.4117 | 0.5669 | 0.0749 | 0.0603 | 0.0798 |
| lfcbm | 4_5 | 0.4269 | 0.5839 | 0.1086 | 0.0930 | 0.0962 |
| lfcbm | 4_8 | 0.4373 | 0.5952 | 0.1551 | 0.1372 | 0.1129 |
| lfcbm | 4_12 | 0.4413 | 0.5997 | 0.2035 | 0.1879 | 0.1262 |
| lfcbm | 4_15 | 0.4438 | 0.6012 | 0.2254 | 0.2183 | 0.1320 |
| lfcbm | 5_3 | 0.4114 | 0.5687 | 0.0863 | 0.0633 | 0.0912 |
| lfcbm | 5_5 | 0.4374 | 0.5962 | 0.1209 | 0.0960 | 0.1026 |
| lfcbm | 5_8 | 0.4421 | 0.6016 | 0.1677 | 0.1403 | 0.1157 |
| lfcbm | 5_12 | 0.4484 | 0.6083 | 0.2154 | 0.1909 | 0.1273 |
| lfcbm | 5_15 | 0.4597 | 0.6183 | 0.2363 | 0.2211 | 0.1325 |

Table 9: Results on cbm in cub200 dataset with unsupervise concepts.

| models | Prompt_Topk | ConceptScore | Ref-ConceptScore | BLUE | METEOR | ROUGE |
|--------|-------------|--------------|------------------|------|--------|-------|
| lfcbm | 1_3 | 0.4433 | 0.5855 | 0.0383 | 0.0195 | 0.0583 |
| lfcbm | 1_5 | 0.4533 | 0.6007 | 0.0607 | 0.0252 | 0.0667 |
| lfcbm | 1_8 | 0.4603 | 0.6132 | 0.0874 | 0.0330 | 0.0753 |
| lfcbm | 1_12 | 0.4672 | 0.6211 | 0.1068 | 0.0420 | 0.0816 |
| lfcbm | 1_15 | 0.4681 | 0.6225 | 0.1093 | 0.0480 | 0.0839 |
| lfcbm | 2_3 | 0.4244 | 0.5662 | 0.0415 | 0.0174 | 0.0657 |
| lfcbm | 2_5 | 0.4401 | 0.5835 | 0.0651 | 0.0232 | 0.0705 |
| lfcbm | 2_8 | 0.4472 | 0.5995 | 0.0896 | 0.0311 | 0.0757 |
| lfcbm | 2_12 | 0.4613 | 0.6150 | 0.1018 | 0.0403 | 0.0793 |
| lfcbm | 2_15 | 0.4625 | 0.6165 | 0.0974 | 0.0464 | 0.0805 |
| lfcbm | 3_3 | 0.4484 | 0.5980 | 0.0700 | 0.025 | 0.0749 |
| lfcbm | 3_5 | 0.4544 | 0.6068 | 0.0896 | 0.0307 | 0.0774 |
| lfcbm | 3_8 | 0.4701 | 0.6232 | 0.1093 | 0.0384 | 0.0801 |
| lfcbm | 3_12 | 0.4765 | 0.6299 | 0.1188 | 0.0478 | 0.0826 |
| lfcbm | 3_15 | 0.4787 | 0.6323 | 0.1146 | 0.0538 | 0.0838 |
| lfcbm | 4_3 | 0.4559 | 0.6005 | 0.0624 | 0.0198 | 0.0537 |
| lfcbm | 4_5 | 0.4609 | 0.6104 | 0.0796 | 0.0255 | 0.0618 |
| lfcbm | 4_8 | 0.4792 | 0.6296 | 0.0976 | 0.0334 | 0.0703 |
| lfcbm | 4_12 | 0.4843 | 0.6353 | 0.1065 | 0.0430 | 0.0767 |
| lfcbm | 4_15 | 0.4863 | 0.6376 | 0.1030 | 0.0491 | 0.0793 |
| lfcbm | 5_3 | 0.4495 | 0.6009 | 0.0738 | 0.0227 | 0.0734 |
| lfcbm | 5_5 | 0.4658 | 0.6182 | 0.0916 | 0.0286 | 0.0759 |
| lfcbm | 5_8 | 0.4755 | 0.6285 | 0.1093 | 0.0369 | 0.0787 |
| lfcbm | 5_12 | 0.4819 | 0.6351 | 0.1167 | 0.0467 | 0.0814 |
| lfcbm | 5_15 | 0.4842 | 0.6376 | 0.1116 | 0.0529 | 0.0826 |

Table 10: Results on phcbm in cub200 dataset.

| models | Prompt_Topk | ConceptScore | Ref-ConceptScore | BLUE | METEOR | ROUGE |
|--------|-------------|--------------|------------------|------|--------|-------|
| phcbm | 1_3 | 0.3867 | 0.5374 | 0.0574 | 0.0667 | 0.1306 |
| phcbm | 1_5 | 0.3951 | 0.5531 | 0.1057 | 0.1039 | 0.1618 |
| phcbm | 1_8 | 0.4169 | 0.5775 | 0.1883 | 0.1573 | 0.1935 |
| phcbm | 1_12 | 0.4217 | 0.5831 | 0.2917 | 0.2221 | 0.2196 |
| phcbm | 1_15 | 0.4348 | 0.5950 | 0.3555 | 0.2668 | 0.2318 |
| phcbm | 2_3 | 0.3770 | 0.5275 | 0.0642 | 0.0646 | 0.1294 |
| phcbm | 2_5 | 0.3889 | 0.5429 | 0.1201 | 0.1016 | 0.1576 |
| phcbm | 2_8 | 0.4013 | 0.5616 | 0.2076 | 0.1542 | 0.1850 |
| phcbm | 2_12 | 0.4078 | 0.5688 | 0.3033 | 0.2174 | 0.2059 |
| phcbm | 2_15 | 0.4123 | 0.5735 | 0.3510 | 0.2607 | 0.2147 |
| phcbm | 3_3 | 0.4036 | 0.5616 | 0.1045 | 0.0717 | 0.1251 |
| phcbm | 3_5 | 0.4196 | 0.5796 | 0.1570 | 0.1085 | 0.1521 |
| phcbm | 3_8 | 0.4314 | 0.5925 | 0.2359 | 0.1608 | 0.1818 |
| phcbm | 3_12 | 0.4372 | 0.5992 | 0.3264 | 0.2239 | 0.2072 |
| phcbm | 3_15 | 0.4441 | 0.6058 | 0.3778 | 0.2666 | 0.2193 |
| phcbm | 4_3 | 0.4036 | 0.5599 | 0.1009 | 0.0670 | 0.1197 |
| phcbm | 4_5 | 0.4127 | 0.5710 | 0.1517 | 0.1044 | 0.1490 |
| phcbm | 4_8 | 0.4302 | 0.5901 | 0.2280 | 0.1578 | 0.1793 |
| phcbm | 4_12 | 0.4345 | 0.5952 | 0.3153 | 0.2224 | 0.2048 |
| phcbm | 4_15 | 0.4358 | 0.5964 | 0.3646 | 0.2669 | 0.2170 |
| phcbm | 5_3 | 0.4034 | 0.5617 | 0.1126 | 0.0701 | 0.1226 |
| phcbm | 5_5 | 0.4290 | 0.5893 | 0.1643 | 0.1075 | 0.1492 |
| phcbm | 5_8 | 0.4348 | 0.5960 | 0.2409 | 0.1608 | 0.1785 |
| phcbm | 5_12 | 0.4409 | 0.6029 | 0.3271 | 0.2250 | 0.2037 |
| phcbm | 5_15 | 0.4486 | 0.6102 | 0.3751 | 0.2689 | 0.2159 |

Table 11: ConceptScores of lfcbm on cifar datasets.

| models | Prompt_Topk | cifar10 | cifar100 |
|--------|-------------|---------|----------|
| lfcbm | 1_3 | 0.4690 | 0.4864 |
| lfcbm | 1_5 | 0.4767 | 0.4911 |
| lfcbm | 1_8 | 0.4797 | 0.4861 |
| lfcbm | 1_12 | 0.4796 | 0.4831 |
| lfcbm | 1_15 | 0.4816 | 0.4830 |
| lfcbm | 2_3 | 0.4328 | 0.4483 |
| lfcbm | 2_5 | 0.4440 | 0.4571 |
| lfcbm | 2_8 | 0.4509 | 0.4617 |
| lfcbm | 2_12 | 0.4536 | 0.4601 |
| lfcbm | 2_15 | 0.4609 | 0.4627 |
| lfcbm | 3_3 | 0.4806 | 0.4910 |
| lfcbm | 3_5 | 0.4782 | 0.4841 |
| lfcbm | 3_8 | 0.4740 | 0.4816 |
| lfcbm | 3_12 | 0.4822 | 0.4857 |
| lfcbm | 3_15 | 0.4824 | 0.4849 |
| lfcbm | 4_3 | 0.4832 | 0.5009 |
| lfcbm | 4_5 | 0.4827 | 0.4934 |
| lfcbm | 4_8 | 0.4778 | 0.4880 |
| lfcbm | 4_12 | 0.4863 | 0.4926 |
| lfcbm | 4_15 | 0.4855 | 0.4914 |
| lfcbm | 5_3 | 0.4815 | 0.4890 |
| lfcbm | 5_5 | 0.4793 | 0.4863 |
| lfcbm | 5_8 | 0.4819 | 0.4896 |
| lfcbm | 5_12 | 0.4873 | 0.4902 |
| lfcbm | 5_15 | 0.4872 | 0.4895 |

Table 12: ConceptScores of phcbm on cifar datasets.

| models | Prompt_Topk | cifar10 | cifar100 |
|--------|-------------|---------|----------|
| phcbm | 1_3 | 0.4531 | 0.4554 |
| phcbm | 1_5 | 0.4623 | 0.4610 |
| phcbm | 1_8 | 0.4676 | 0.4628 |
| phcbm | 1_12 | 0.4661 | 0.4642 |
| phcbm | 1_15 | 0.4740 | 0.4663 |
| phcbm | 2_3 | 0.4088 | 0.4186 |
| phcbm | 2_5 | 0.4199 | 0.4252 |
| phcbm | 2_8 | 0.4287 | 0.4327 |
| phcbm | 2_12 | 0.4343 | 0.4359 |
| phcbm | 2_15 | 0.4398 | 0.4446 |
| phcbm | 3_3 | 0.4681 | 0.4637 |
| phcbm | 3_5 | 0.4668 | 0.4601 |
| phcbm | 3_8 | 0.4637 | 0.4598 |
| phcbm | 3_12 | 0.4755 | 0.4681 |
| phcbm | 3_15 | 0.4769 | 0.4689 |
| phcbm | 4_3 | 0.4731 | 0.4696 |
| phcbm | 4_5 | 0.4728 | 0.4663 |
| phcbm | 4_8 | 0.4677 | 0.4623 |
| phcbm | 4_12 | 0.4788 | 0.4712 |
| phcbm | 4_15 | 0.4783 | 0.4711 |
| phcbm | 5_3 | 0.4698 | 0.4647 |
| phcbm | 5_5 | 0.4701 | 0.4641 |
| phcbm | 5_8 | 0.4710 | 0.4687 |
| phcbm | 5_12 | 0.4822 | 0.4742 |
| phcbm | 5_15 | 0.4835 | 0.4753 |

