# OpenReview forum: "Evaluating the Unseen: A Novel Framework for Assessing Unsupervised Concept Bottleneck Models"
_ICLR.cc/2025/Conference — ICLR 2025 Conference Withdrawn Submission_

### Official Review · Reviewer_9Y9p · 2024-10-31

**Soundness:** 2
**Presentation:** 3
**Contribution:** 1
**Rating:** 3
**Confidence:** 4

**Summary:**

This paper introduces a novel evaluation framework for assessing the quality of explanations generated by unsupervised concept bottleneck models (CBMs). It addresses the challenge of evaluating unsupervised and label-free CBMs, which, without ground-truth labels, are difficult to assess in terms of reliability and accuracy. The proposed metrics, including ConceptScore, Ref-ConceptScore, and some NLP metrics, focus on measuring relevance, consistency, and informativeness. Experiments validate the framework’s effectiveness, showing alignment between the framework’s results, large language model (LLM) evaluations, and human judgments.

**Strengths:**

- This paper presents a novel framework for evaluating unsupervised concept learning, addressing an important gap not tackled by existing works.
- The two newly proposed evaluation metrics, ConceptScore and Ref-ConceptScore, leverage the LongCLIP model to assess the alignment between an image and a concept. This innovative approach introduces a reasonable and effective way to gauge the quality of unsupervised concept learning.
- The experiments are extensive and include comparisons with both human annotations on labeled datasets and GPT-generated outputs, offering a thorough validation of the proposed framework.
- The paper is well-written, with clear motivation and explanations, making it easy for readers to follow and understand the significance of the contributions.

**Weaknesses:**

- I believe the novelty of this paper is very limited, as it primarily modifies existing cosine similarity-based metrics using a pre-existing LongCLIP model developed in another paper.
- While this paper proposes some new metrics, they are still based on the existing pretrained LongCLIP model, and using GPT-based evaluation alone to demonstrate effectiveness may not be sufficient for unlabeled datasets.
- The NLP-based metrics section is trivial and can be moved to Appendix.
- In the correlation analysis section, the correlation scores appear to be low, which does not adequately demonstrate the effectiveness and faithfulness of the proposed metrics.

**Questions:**

- Could you clarify how this paper’s novelty extends beyond modifying existing cosine similarity-based metrics with the pre-existing LongCLIP model? Given that the new metrics rely heavily on this pretrained model, how does your work build on or differentiate itself from the GPT-based approaches in the real-wold application (as you use correlation between your proposed metrics and GPT to demonstrate)?
- In the correlation analysis, the observed correlation scores appear relatively low. Could you elaborate on how these scores support the effectiveness and faithfulness of the proposed metrics, how to distinguish what score denotes good and what denotes not good, and are there ways to improve alignment with human judgments?

---

### Official Review · Reviewer_1RpX · 2024-11-02

**Soundness:** 2
**Presentation:** 2
**Contribution:** 2
**Rating:** 3
**Confidence:** 4

**Summary:**

The paper introduces a novel evaluation framework for unsupervised Concept Bottleneck Models (CBMs). The framework addresses the challenge of assessing the quality and relevance of concepts generated by unsupervised CBMs without ground-truth labels. The authors propose several metrics, including ConceptScore and Ref-ConceptScore, which leverage the Long-CLIP model to measure the semantic coherence between predicted concepts and data points. The paper validates these metrics through experiments on CIFAR-10, CIFAR-100, and CUB-200 datasets, demonstrating positive correlations between the proposed metrics, human judgments, and GPT evaluations.

**Strengths:**

**Paper presentation and structure**: the paper is well written and structured, easy to follow. It presents the paper key contributions already from the introduction in a clear way. The rest of the paper is similarly presented, although the method could benefit of more examples and better definitions of some terms in the method section.

**Weaknesses:**

## Major Issues

- **Framework positioning & evaluated methods**: one of the main issues of the paper in its current form is the positioning with respect to the literature. While the paper clearly position itself for non-supervised methods, the related work and the experiments review and employ i) supervised methods that either the train concept layer together with the rest of the network (CBM, CEM) or after (Post-hoc CBM), ii) LLM-based CBMs utilizing VLMs to provide concept annotations. No actual unsupervised methods are reported such as [1,2,3,4]. Thus, I only see two possibilities: either the authors reframe the proposed framework to evaluate supervised or LLM-based concept bottleneck models, or they integrate actual unsupervised CBM models in the evaluation and in the related work.
- **Related work**: while providing novel interesting metrics for studying and evaluating CBMs, the authors should also tell thy these metrics are required and why existing metrics are not sufficient to represent the same information. Among other papers, the authors could take a look at [5] which provides several metrics to evaluate concept representations for both supervised and unsupervised models.
- **Result presentation**: The result presentation is very confusing and below the standard of the conference. Prompt1 is never defined. Several results are report within the text without specifying for which model they have been computed. No standard deviation is provided.

[1] Dai, Enyan, and Suhang Wang. "Towards self-explainable graph neural network." Proceedings of the 30th ACM International Conference on Information & Knowledge Management. 2021.

[2] Wang, Bowen, et al. "Learning bottleneck concepts in image classification." Proceedings of the ieee/cvf conference on computer vision and pattern recognition. 2023.

[3] Chen, Chaofan, et al. "This looks like that: deep learning for interpretable image recognition." Advances in neural information processing systems 32 (2019).

[4] Rajagopal, Dheeraj, et al. "SELFEXPLAIN: A Self-Explaining Architecture for Neural Text Classifiers." Proceedings of the 2021 Conference on Empirical Methods in Natural Language Processing. 2021.

[5] Zarlenga, Mateo Espinosa, et al. "Towards robust metrics for concept representation evaluation." Proceedings of the AAAI Conference on Artificial Intelligence. Vol. 37. No. 10. 2023.


## Minor Issues
- Figure 1 is not very clear, and its caption does not help in understanding it.
- In Eq.1 there are no parenthesis that define the arguments of the cos similarity
- Section 3.3, “The harmonic mean is particularly useful when dealing with ratios or rates, as it gives more weight to lower values, ensuring that a single low ConceptScore does not dominate the overall evaluation”. It is not very clear: as it gives more weight to lower values, a single low ConceptScore should dominate the overall evaluation, right?
- Ref-Concept score: it is not clear the notation used to define the concept ground truth in Equation 2-3. In general the way in which the concepts are encoded in a prompt is unclear, an example would have improved the understanding. It is also unclear while there are 2 equations.
- It is not clear in section 3.4 how the author consider concept predictions and labels in the NLP contexts.
- Section 3.5 is very badly written: the human score is not defined at all, there is no indication of the actual rating provided by humans nor how they should assign it.
- No standard deviation reported for any metric in the experiments.

**Questions:**

- Ref-Concept score, in Equation 2-3: why is the concept ground truth also passed through the text encoder E_t? How does it differ from the “predicted concepts”?
- Can you provide some examples of how you consider concept prediction and labels as text?

---

### Official Review · Reviewer_Fygi · 2024-11-02

**Soundness:** 1
**Presentation:** 3
**Contribution:** 2
**Rating:** 1
**Confidence:** 4

**Summary:**

The authors evaluate textual concepts extracted by concept bottleneck models using several measures. One measure uses directly a similarity using a vision-text embedding between the image and the predicted concepts. Another measure uses the first measure in a harmonic mean together with a embedding space similarity between predicted and ground truth concepts.  They also employ BLEU, METEOR and Rouge between predicted and ground truth concepts, and finally they use humans and a GPT family model to rate the alignment between the image and textual concepts.

**Strengths:**

They measure an aligment between images and predicted concepts using external vision language models. They perform measurements between predicted and ground truth concepts.

**Weaknesses:**

This paper suffers from a conceptual fallacy:

it states to analyze the usefulness of extracted concepts for a prediction task (see below the abstract states) which relies on a concatenation of two mappings, g and f (lines 150-154), but the proposed evaluation methods do not use in any way the mapping f which performs the prediction task.

It makes the assumption that LLMs and humans would use the extracted/predicted concepts c in the same way as f does (or that longclip acts similar to g). That is the fallacy.

If f performs a low level vision task, like extracting parts with unusual optical flow, or parts with adversarial statistics, then it will not use the same high level semantics as longclip, gpt or humans.
Other examples can be made, also with divergent high level semantics, for example when creating synthetised reasoning tasks, or more concretely a movie where a bird is symbolizing upcoming demise and thus is labeled correctly as "yellow doombringer".

In a high level summary, the paper measures aspects of alignment between the extracted concepts and longclip/ LLMs / humans, but a low score does not indicate any lack of usefulness or importance of any sort for the mapping f(g(x)).

the abstract states:

"This paper introduces a comprehensive evaluation framework
designed to assess the quality of explanations produced by unsupervised CBMs."

"Our framework comprises a set of novel metrics that evaluate various aspects of
the concept outputs, including their relevance, consistency, and informativeness."

The quality is not measured with respect to the mapping for which they are used.
The relevance and informativeness is not measured with respect to the mapping for which they are used.


contribution 1 in line 073/074 states:  "We propose a comprehensive evaluation framework for unsupervised CBMs, incorporating a range of metrics tailored to assess concept quality from multiple angles."

The evaluation does not measure concept quality with respect to the task at hand and the trained mapping at hand.

The paper cannot be accepted without completely removing such misleading claims.
As such the paper is a clear reject.

The above big issue is the reason for a very low score in soundness.


The NLP evaluation makes somewhat sense, however similarities on word level between predicted and ground truth concepts are not very novel and straightforward if one has ground truth concepts available.

minor points:

Concretely, eq. 1 relies on a similarity between the image and the concepts as measured by longclip. That might work out for some tasks with similar semantics like fine grained classification, but there is no step taken to align eq1 to the mapping g. No measurement is done whether g acts in any way similar to Longclip.

eq. 2 has a flaw as well:
eq. 2 uses a harmonic mean between two similarities. One of them is computed from a similarity between embedded predicted and ground truth concepts. The harmonic mean is dominated by lower scores. That means, that if the predicted textual concept is close to the ground truth concept, but the similarity to the image is low, one will get a low score, while one should get a high score instead.

**Questions:**

NA

---

### Official Review · Reviewer_akse · 2024-11-03

**Soundness:** 3
**Presentation:** 3
**Contribution:** 3
**Rating:** 5
**Confidence:** 3

**Summary:**

This paper proposes an evaluation framework for the assessment of the quality of explanations produced by unsupervised concept bottleneck models. The approach allows to evaluate different aspects of the concept outputs, including their relevance, consistency, and informativeness. Experiments are conducted on different models and correlations between between the scores derived with the framework and LLM evaluations and human judgments are analysed.

**Strengths:**

The paper addresses a timely topic, namely the assessment of the quality of explanations produced by unsupervised CBMs. While the proposed ConceptScore represents a relatively simple approach based on cosine similarities, the Ref-ConceptScore approach is a more elaborated metric, which allows to incorporate ground truth concept annotations when available.
The paper is clearly written and the appendix contains very helpful materials. The technical originality and quality of the proposed metric is rather incremental. The experimental comparison between the derived scores, LLM and human is very interesting and represent the main result of the paper.

**Weaknesses:**

The technical contribution is limited. The approaches utilises models such as CLIP or LongCLIP and proposes to use simple metrics based on cosine similarity or more complex variants thereof. While the experimental comparison between the derived scores, LLM and human is in itself very interesting, the paper lacks more in-depth investigations and experiments showing the practical impact of the proposed approach. I understand that the method is only a starting point "for further research into more transparent and trustworthy AI systems", but still having some experiments demonstrating the practical impact and actionability of the approach would be good.

**Questions:**

What are the practical impacts of this work?
How could the framework contribute in building more transparent and trustworthy AI?

---

### Note · Authors · 2024-11-22

I have read and agree with the venue's withdrawal policy on behalf of myself and my co-authors.